# The Analysis Effect of Selected Factors on the Shear Strength of Woodbark at Different Wood Species

Peter Vilkovský [1,*], Tatiana Vilkovská [1], Ivan Klement [1] and Igor Čunderlík [2]

1 Department of Wood Technology, Faculty of Wood Sciences and Technology, Technical University in Zvolen, T. G. Masaryka 24, 96001 Zvolen, Slovakia; t.hurakova21@gmail.com (T.V.); klement@tuzvo.sk (I.K.)
2 Department of Wood Science, Faculty of Wood Sciences and Technology, Technical University in Zvolen, T. G. Masaryka 24, 96001 Zvolen, Slovakia; i.cunderlik@gmail.com
* Correspondence: peter.vilkovsky@tuzvo.sk

**Abstract:** The bark as a product of the dividing of wood and cork cambium consists of a set of protective layers of cells, which protect the living tissue (cambium) from the external environment and separate the bark from the wood. The structure of bark as a component of a living tree is completely different from wood. This article describes the testing of the adhesion of wood/bark in the longitudinal and tangential anatomical direction during the dormant and growing season on three choice wood species (oak, beech, and spruce). The results show a remarkable influence of the wood species and anatomical direction, as well as period of vegetation (dormant or growing season). All wood species had higher values of shear strength in the longitudinal direction compared to the tangential direction. The highest average values in the longitudinal direction were measured in the dormant period for sessile oak (0.49 MPa) and beech (0.48 MPa). The lowest value of shear strength in the longitudinal direction was measured for spruce (0.36 MPa). During the growing season, the highest average shear strength values were also measured in the longitudinal direction at beech (0.46 MPa) and oak (0.39 MPa). The lowest value of shear strength in the longitudinal direction was measured similarly for spruce (0.26 MPa).

**Keywords:** shear strength; beech; oak; bark; cambium; spruce

## 1. Introduction

The bark conceals a structurally more complex composition compared to wood as a result of the division of wood and cork cambium. The key functions of bark include protection of plant stems, both through the physical and chemical nature of the rhytidome, and the responsive activity of the parenchyma (living cells) located in the inner bark [1]. The structure of the bark consists of all the outer layers of cambium, which are divided into the tissues of the un-collapsed and collapsed phloem and periderm. Secondary phloem, also called the inner bark, which contains tissues created by cambial division, is the layer closest to the cambium [2]. Similarly, the authors of [3] stated that bark is composed of inner bark and outer bark layers forming a morphological continuum with wood that is found on the inner side of the cambium zone. According to [4], the inner bark is a layer that is spread from the cambium to the last formed or innermost layer of the periderm. Periderm, called the outer bark, is formed by a layer of meristematic tissues of phelloderm, phellogen (cork cambium), and phellem (cork) [5–8]. The wood/bark adhesion interface is of great importance from the point of view of the protection of living tissues on trunk, branches, and roots. On the other hand, wood/bark adhesion plays an important role in the debarking of logs, especially in the pulp production processes [9,10]. Adhesion is influenced by several factors, especially the moisture of the wood and the bark, the period of growth, and the structure of the cambium, but also the structure of the layers close to it and especially the inner bark (early and late phloem and phloem rays) [9,11–13]. The structure of these layers depends on the type of wood, from which it can be expected that the wood/bark adhesion may also

differ depending on the type of wood. Differences in the structure of the inner bark can be found, for example, in the bark tissues of the winter oak *(Quercus petraea* Liebl.), in which the bark layer consists of sieve tubes, accompanying cells, and phloem fibers, as well as the axial and ray parenchyma [14]. Oak generally contains phloem rays that are broad, lignified, and accompanied by sclerified cells contains of crystals (sclereids). In close proximity to the cambium, we can observe 1–3 rows of parenchymal cells [14]. In contrast, the inner bark of Silver fir (*Abies alba* Mill.) and Norway spruce (*Picea abies* L., Karst.) is relatively simple and consists of living vertically oriented sieve tubes and various types of parenchyma cells. The ray system in Silver fir (*Abies alba* Mill.) contains single-row rays, and in Norway spruce (*Picea abies* L., Karst.) single-row to three-row rays. Parenchyma groups in phloem separate early phloem from late phloem. Between them, the sieve tubes in larger or smaller tangential groups (usually 1 or 2 cells) are allocated [15]. The beech tree (*Fagus sylvatica* L.), whose structure of the inner bark, as well as the peridermis, has also been described in detail in the works [6,16–18], is also characterized by a significantly different bark. According to [17], beech bark tissue consists of collapsed and uncollected phloem and periderm. The cambium, a layer composed of highly vacuumized meristematic cells arranged radially, can have the remarkable effect on wood/bark adhesion. Unlike other meristems, cambium is a complex of tissues containing two morphologically distinct cell types: axially elongated spindle cambial cells and isodiametric beam cambial cells [19]. According to [18–20], activity of cambium can remarkably influence the of choice mechanical properties (for example shear strength) during the dormant and vegetation period. The research in [10] discovered that harvesting time had a significant effect on wood/bark strength of shrub willow ($p < 0.01$). Shrub willow (*Salix* sp.) harvested during the winter had significantly ($p < 0.01$) higher wood/bark strength compared to early-growing, with a mean increase of 90% and 108% compared to mid-year. This is thought to be due to seasonal changes in cambium layer morphology. The authors of [21] described differences in cambium during the dormant and vegetation periods and found that cambium cell walls in the dormant period have a denser fibril network, smaller pore size, and fewer cross-links between fibrils than active cambium cell walls. According to the results of [22], a significant influence of the bark thickness on the cambium activity can also be expected. This assumption was based on measurement of older winter oak with a rough bark, which prevent penetration heat and, thus, did not affect cambial activity. In the case of young winter oak, which had a thin bark, the effect was reversed and the cambial tissues were very active. Therefore, overall, these findings showed that several factors act simultaneously on cambial activity (bark thickness, environmental temperature, etc.). The evaluation of wood/bark adhesion was studied by several authors [9–13,23,24], while different wood species were tested, which indicated different values of shear strength between the tested trees, but also between the dormant and vegetation period. The authors of [11] tested wood/bark adhesion during both the growing and dormant periods. During testing, they pointed mainly to seasonal changes affecting the cambial zone, which were the cause of different values of shear strength. Dormant cambium showed a lower proportion of cambial cells compared to active cambium. For example, the cambial zone in the oak contained 5 to 6 cells radially, and, conversely, in the maple, the cambial zone consisted of 6 to 8 cells in the radial direction during dormant season. During the growing season, active cambium began to expand the thickness of the cambial zone, which ranged from about 20 cells in oak and 10 to 12 cells in maple. During testing wood/bark adhesion in the dormant period, the disruption occurred mainly in the inner bark (phloem), most often in the area between the tangential bands of sclerified fibers (oak) or sclereids (maple) and the bands of parenchyma cells connected to adjacent sieve tubes. The authors of [12] focused on testing wood/bark adhesion on deciduous and coniferous North American trees during the growing and dormant periods. Tree species tested were oak, spruce, poplar, birch, and others (Figure 1). All selected wood species showed higher shear strength in the dormant period than in the vegetation period. Disruption usually occurred in the sieve tubes and parenchyma cells of the inner bark. Dormant period and higher shear strength were often associated with the presence of a

large number of bast fibers and rays, which led to the assumption that phloem fibers may be one of the reasons for increasing wood/bark adhesion.

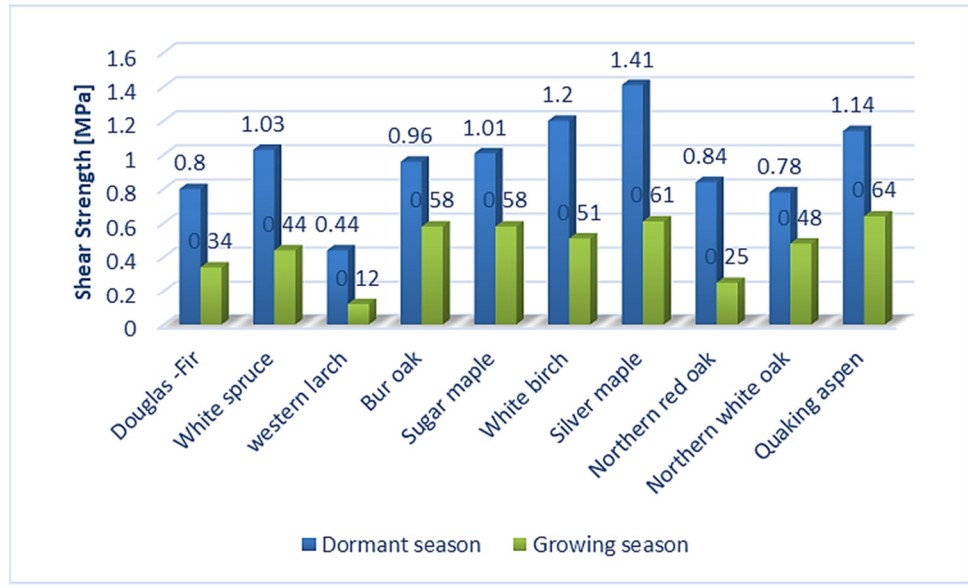

**Figure 1.** The values of shear strength in longitudinal direction during the growing season and dormant period on selected tree species [12].

The authors of [25] studied the effects of temperature and moisture content on bark/wood shear strength. Fifteen stems of black spruce (*Picea mariana* Mill.) and balsam fir (*Abies balsamea* (L.) Mill.) logs were selected and crosscut into three 1.25 m log sections. Shear strength was measured at five temperatures, ranging from 10 to −30 °C, and at five levels of sapwood moisture contents. Their results showed that temperature had a significant effect on shear strength for both species. In addition, results showed that black spruce had higher BWSS than balsam fir. Mean values shear strength at −30 °C varied from 2.20 to 1.53 MPa in black spruce and between 1.99 and 0.79 MPa in balsam fir. Mean values shear strength at 10 °C varied from 0.53 to 0.71 MPa in black spruce and between 0.56 and 0.57 MPa in balsam fir.

The aim of the present research was to analyze the influence of wood species and anatomical direction of testing on wood/bark adhesion. Testing was performed in two anatomical directions, for three selected wood species, which differed mainly in the type of bark, thickness increase of bark tissues, and in the total proportion of bark. The measurements were performed in February and May, i.e., during the dormant and vegetation period. The results can be of great benefit not only for a better understanding of the protective function of the bark, but also for the debarking process, in which the bark is separated from the wood in the tangential direction.

## 2. Material and Method

The samples of beech (*Fagus sylvatica* L.), Sessile oak (*Quercus petraea* L.), and spruce (*Picea abies* M.) were obtained at the University Forest Enterprise of the Technical University in Zvolen (Slovakia) from the Včelien locality. Location for sampling was in the Kremnické vrchy mountains at an altitude of about 550 m.a.s.l. Samples were taken from a trunk height of about 130 cm (Figure 2A), during the dormant period (February) and the growing period (May). Two samples about 5 cm thick were obtained from each log. One sample was used to prepare test specimens tested in the longitudinal direction (parallel to the fibers direction), and the other paired sample was prepared to make test specimens in the tangential direction (across to the fibers direction). Samples were tested in fresh state.

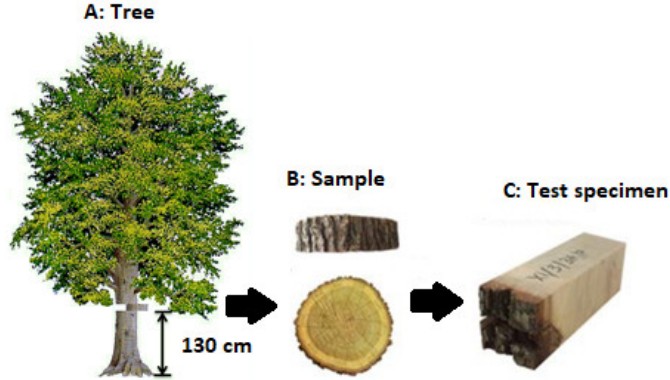

**Figure 2.** Sequential steps to get the samples for testing the shear strength.

Ten test specimens of dimensions 30 × 30 × 50 (T × R × L) were made from each sample. Dimensions of the specimens were adapted for the hole of the anchor and the jaw edge. Wood/bark adhesion testing, through shear strength, was performed using the TIRA TEST 2150 equipment. The loading force was transferred through a specially shaped jaw to the bark of the test specimens along or across to the fibers' direction (Figure 3). Movement of jaw and intensity of the loading force were recorded by ALMEMO 2690 through the output voltage.

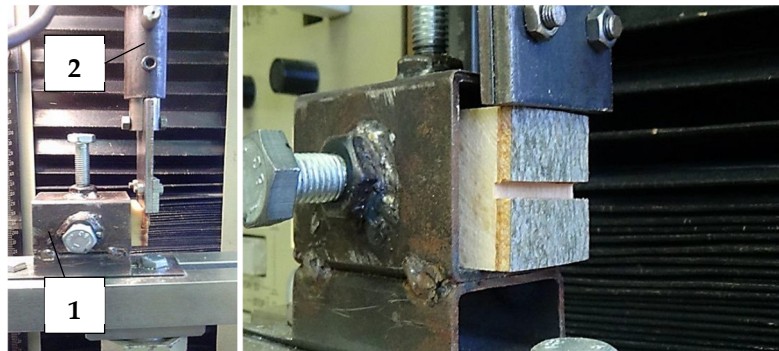

**Figure 3.** Depiction of the samples anchored by screws (1) and the jaw edges on the different wood/bark interfaces (2).

The data together with the shear area dimensions (tangential × longitudinal dimension) were used to calculate the shear strength according to [26]:

$$\tau_{T,L} = \frac{F_{max}}{S} \ [\text{MPa}] \tag{1}$$

where $\tau_{T,L}$ is the limit of the shear strength in the longitudinal (*L*) and tangential (*T*) direction [MPa], $F_{max}$ is the maximum loading force [N], and *S* is the shear area [mm$^2$].

All shear strength values were also processed at the level elementary and inductive statistics. The method of analysis of variance (ANOVA) was used for this statistical evaluation.

## 3. Results and Discussion

Wood/bark adhesion evaluated through the shear strength reached higher values in the dormant period for all tested wood species compared to the shear strength values measured during the growing season. This indicates a remarkable effect of the growing season on the shear strength value at the interface wood/bark. Similar differences between the dormant and vegetation season were obtained by other authors [13,18], who attributed higher values measured during the dormant period to a large number of phloem fibers and rays or a lower proportion cambial cells compared to the vegetation season. The

results also confirmed a remarkable influence of the anatomical direction in all tested wood species (Figure 4). All wood species had higher values of shear strength in the longitudinal direction compared to the tangential direction. The highest average values in the longitudinal direction were measured in the dormant period for sessile oak (0.49 MPa) and beech (0.48 MPa). The lowest value of shear strength in the longitudinal direction was measured for spruce (0.36 MPa). The same was noted in the dormant period, even in the research of [10] who, among other wood species, tested oak (*Quercus alba*) and spruce (*Picea glauca*) (Figure 5). In the dormant season, they measured the average value of shear strength in the longitudinal direction for spruce (Picea glauca), 1.03 MPa, and in the vegetation season, 0.44 MPa, and for oak (*Quercus alba*), they measured 0.78 MPa in the dormant period and 0.48 MPa in the vegetation period. This assertion was also confirmed by measurements of the shear strength in [9].

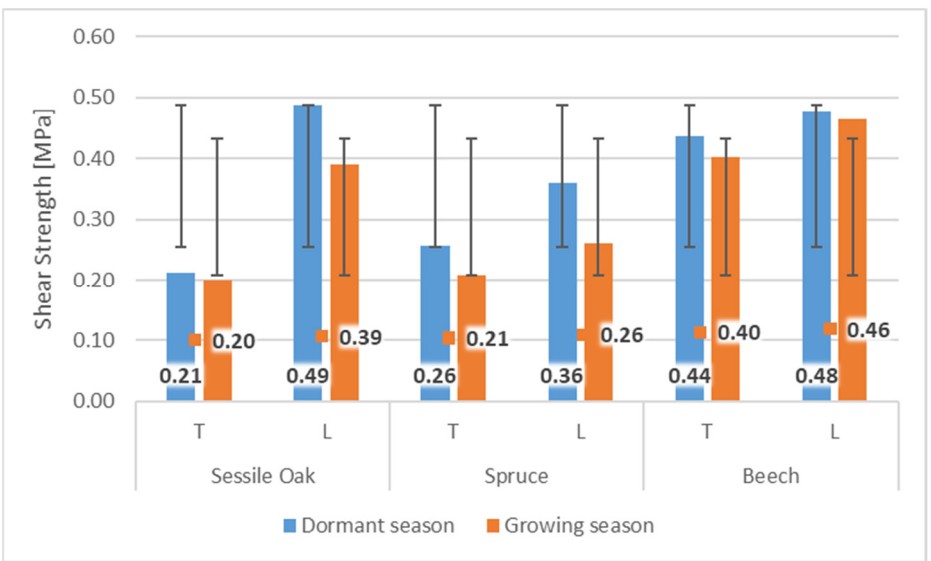

**Figure 4.** The values of shear strength during dormant and growing season in tangential (T) and longitudinal (L) directions for sessile oak, spruce, and beech.

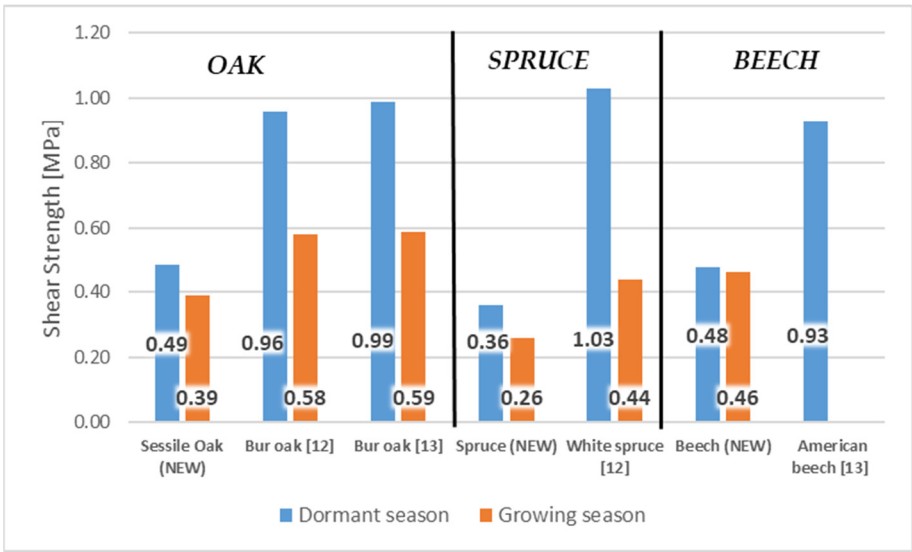

**Figure 5.** Comparison values of shear strength during dormant and growing season in longitudinal direction with similar wood species.

The higher resistance, not only in the dormant season, can often be ascribed even with the large number of phloem fibers and rays that beech (phloem rays) and oak (phloem

fibers and rays) have in their structure. Therefore, we can assume that phloem fibers under load will behave the same as mechanical tissues in wood and will have similarly different mechanical properties in different directions of loading or shear strength values. This was also stated by [14], who examined the structure of the inner bark of the sessile oak (*Quercus petraea Liebl*.), as well as [17], who examined the bark of beech (*Fagus sylvatica* L.). Spruce does not contain a mechanical tissue, which could, remarkably, increase the shear strength. Research by [15] describes the structure of the inner bark of the spruce (Picea abies) as simple and consisting of living vertically oriented sieve tubes and various types of parenchyma cells. During wood/bark adhesion testing, the shear strength in the transverse (tangential) direction lower values were measured and compared to the longitudinal direction. The largest difference in the direction of testing was found for the oak. In the dormant season, the values in the tangential direction were lower compared to the longitudinal direction by more than 57%, and in the growing season by almost 49%. smallest difference was found in beech wood, where in the dormant period the difference was measured at only 8%, and in the growing season 13%. During the growing season, the highest average shear strength values were also measured in the longitudinal direction in beech (0.46 MPa) and oak (0.39 MPa). The lowest value of shear strength in the longitudinal direction was measured similarly for spruce (0.26 MPa). In the tangential direction, remarkable lower values were measured during the growing season compared to the longitudinal direction; oak by almost 49%, beech by almost 20%, and spruce by 13%.

Based on the value of the significance level ($p$ = 0.00000), it was confirmed that the direction of the loading force remarkably affects the resulting shear strength (Figure 6).

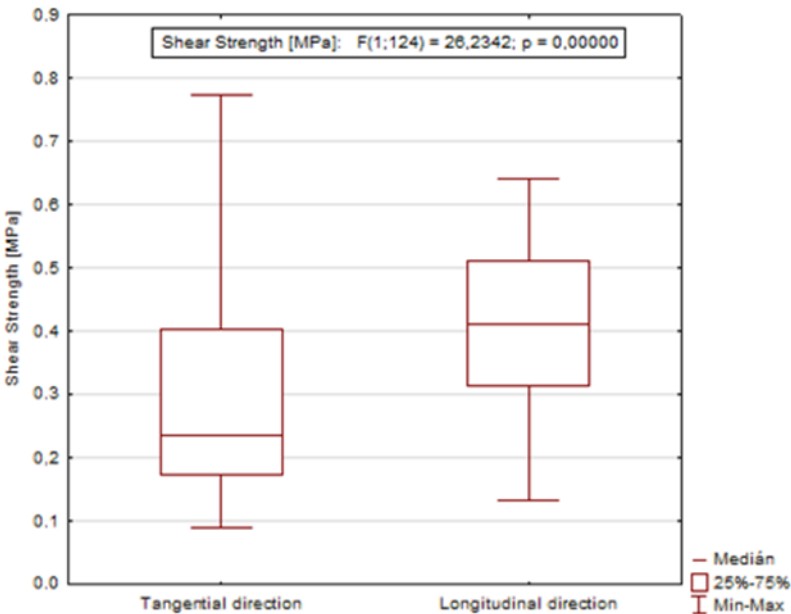

**Figure 6.** Graph showing the dependence of shear strength on the direction of loading during dormant and growing season.

The application of analysis of variance (ANOVA) confirmed another statistically significant factor, namely the wood species that remarkably affects the value of shear strength in the longitudinal and tangential direction (Figure 7). This statement is also confirmed by the value of the significance level $p$ = 0.0000.

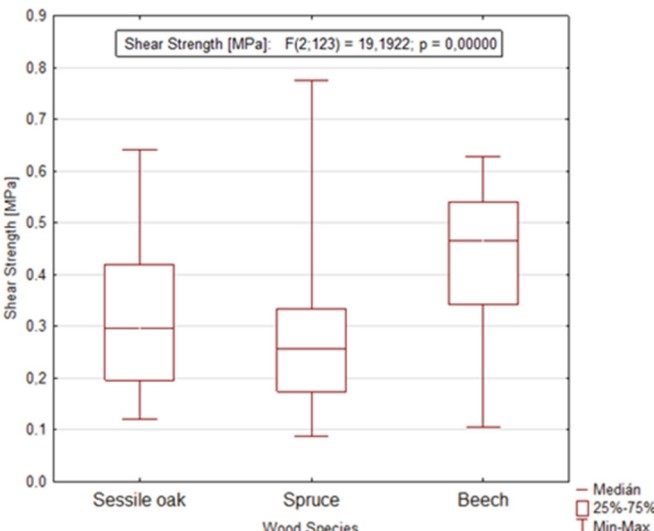

**Figure 7.** Graph showing the dependence of shear strength on the wood species during dormant and growing season in the longitudinal and tangential direction.

The reason for the different shear strength may be the different structure of the bark, and even the different structure of cambium during dormant and growing season. Main differences in structure can be found in the proportion of mechanical tissues that oak bark has in the form of phloem fibers and sclereids, or even in multi-row bast rays. Unlike oak, beech has a significant proportion of sclereids (thick-walled cells) in the bark structure, while bast fibers are absent. Compared to beech and oak, spruce does not contain any mechanical tissues, and single-row to three-row phloem rays are typical for spruce. This statement also corresponds to the works of [12,13,24], which focused on testing of the shear strength of several woods. On the contrary, according to several studies [18–21] examining the cambium layer, it was found that the structure of cambium significantly affects different mechanical properties during the dormant and vegetation season. According to the authors [21], cambium cell walls in the dormant period have a denser network of fibrils, smaller pore size, and a lower number of transverse tissues between fibrils than active cambium cell walls. This may explain the higher shear strength values during the dormant period.

## 4. Conclusions

The aim of this work was to determine the influence of the wood species and the direction of the loading force on the shear strength during the dormant and vegetation period in two anatomical directions, longitudinal and tangential. Three species, namely sessile oak (*Quercus petraea* (Matt.) Liebl.), European beech (*Fagus sylvatica* L.), and Norway spruce (*Picea abies* L. Karst.) were selected for testing. The following conclusions can be drawn from the measured data:

- The measured data confirmed a remarkable influence of the direction of the loading force for all tested wood species;
- The highest difference between the tangential and longitudinal direction was measured in sessile oak. The longitudinal direction had average values of shear strength more than 57% higher than the tangential direction during the dormant season; the longitudinal direction in oak had a value of 0.49 MPa, and the tangential direction had a value of 0.21 MPa;
- The results also showed a remarkable effect of the testing season (growing and dormant). The highest values were measured during the dormant season. The rea-son can be found in the structure of cambium, which differs between the dormant and growing season. The differences are mainly in the quality and quantity of the cambial cells;

- The highest difference between the tangential and longitudinal direction was also measured for sessile oak. The longitudinal direction had average values of shear strength almost 49% higher than the tangential direction during the growing season. The value in the longitudinal direction was 0.39 MPa, and in the tangential direction was 0.20 MPa;
- Minor differences in shear strength values were found for the remaining wood species. The smallest difference was found in beech. During the dormant season, the difference was only 8% and during the vegetation season was 13%;
- Based on the analysis of variance (ANOVA), whose value of the significance level was $p = 0.0000$, a remarkable influence of the wood species on the final value of shear strength was confirmed. This result can be explained mainly by the different structure of the bark between the tested wood species. Mainly, differences in structure can be found in the proportion of mechanical tissues, which oak bark has in the form of phloem fibers and sclereids, or even in several-rows of phloem rays, and vice versa; in the case of beech, these fibers are missing. Unlike oak, beech has a remarkable proportion of sclereids (thick-walled cells) in the structure of the bark. Compared to beech and oak, spruce does not contain any mechanical tissues and only single-row to three-row phloem rays are typical;
- In the final evaluation, we can confirm that values of shear strength are significantly affected by the wood species and period of vegetation, as well as direction of the loading force, but also by other factors. These results can be of great benefit, not only in understanding the protective function of the bark, but also in the debarking process, in which the bark is separated from the wood in the tangential direction.

**Author Contributions:** Conceptualization, P.V. and T.V.; methodology, P.V. and I.Č.; software, P.V. and I.K.; validation P.V. and I.K.; formal analysis, P.V.; investigation P.V.; resources, T.V.; data curation, I.K.; writing—original draft preparation, P.V.; writing—review & editing, T.V.; visualization, I.Č.; supervision, I.K.; project administration, P.V. and I.Č.; funding acquisition, P.V. All authors have read and agreed to the published version of the manuscript.

**Funding:** This research was funded by the Slovak Research and Development Agency under the contract no. APVV-17-0583. This study also received funding from VEGA of the Ministry of Education, Science, Research and Sport of the Slovak Republic (No. 1/0063/22). This publication is the result of the project implementation: Progressive research of performance properties of wood-based materials and products (LignoPro), ITMS: 313011T720 supported by the Operational Programme Integrated Infrastructure (OPII) funded by the ERDF.

**Conflicts of Interest:** The authors declare no conflict of interest.

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
