# Peer review of "The Analysis Effect of Selected Factors on the Shear Strength of Woodbark at Different Wood Species"

_forests, doi:10.3390/f13050637_

Round 1

Reviewer 1 Report

Abstract

In the abstract, authors should refer to the main results with an indication of the values (by number or percentage).The general description is insufficient.

If there is enough space, they can also list the species of wood tested here.

Introduction

Before Fig. 1, inserted into the article, it should be referred to in the text. This reference should not be after Fig.

In the case of publications on trees / wood, it is recommended to provide the following information, such as "oaks (Quercus L. Sp. Pl. 994. 1753)"

The caption under Fig. 1 - reference to the literature, should be unified - ie in brackets [].

Do the authors have the required permission to reproduce a figure (from the journal)?

Material and Method

OK

It would be worth adding a photo of an exemplary damaged sample 

Results and Discussion

Fig. 4. It is worth adding what T and L stand for, it can be added in the figure caption.

It is accepted in the discussion to refer to the results of other researchers, in my opinion it is good to write that the other researchers obtained similar results to describe the general nature of the changes.

I believe that authors should refer to the value of other researchers' results, even if they differ from their results. If they differ, it should be discussed.

The authors can easily improve the readability of the analyzed results by adding charts with the indication of changes in percentages, such information could be included in the abstract.

The work is very interesting and I think that I will refer to it (quoted) in the future, but expressing the main changes in percentages will facilitate this task and make it easier to receive the content conveyed.

I think it is very important.

Conclusions

Long but show important content, they can stay. It is also worth providing numerical values.

References

Relatively small number of literature items. Mainly authors from Slovakia. A very large number of cited author Gričar J. (about 30%). Only 2 references are younger than 5 years old. 90% of publications are much older than 5 years, which is unacceptable for good scientific publications. This shows that the authors have not made an up-to-date diagnosis of the state of knowledge and technology. This needs to be corrected.

Current papers analyzing similar properties are available:

Warguła, Ł.; Wojtkowiak, D.; Kukla, M.; Talaśka, K. Symmetric Nature of Stress Distribution in the Elastic-Plastic Range of Pinus L. Pine Wood Samples Determined Experimentally and Using the Finite Element Method (FEM). Symmetry 2021, 13, 39. https://doi.org/10.3390/sym13010039

Kyzioł, L.; Żuk, D.; Abramczyk, N. Determination of shear stresses in the measurement area of a modified wood sample. REVIEWS ON ADVANCED MATERIALS SCIENCE 2022, 61, 146.

Fathi, Hamidreza, Vahid Nasir, and Siavash Kazemirad. "Prediction of the mechanical properties of wood using guided wave propagation and machine learning." Construction and Building Materials 262 (2020): 120848.

Wang, D., Fu, F., & Lin, L. (2022). Molecular-level characterization of changes in the mechanical properties of wood in response to thermal treatment. Cellulose, 1-12.

Kukla, M., & Warguła, Ł. (2021). Wood-based Boards Mechanical Properties and Their Effects on the Cutting Process during Shredding. BioResources, 16(4), 8006-8021.

Author Response

Dear Reviewer

Thank you for your revisions and suggestions as well. We consider it all and did some changes in article. All changes are highlighted by red color.

Yours sincerely,

Authors

Reviewer 2 Report

The topic of the manuscript is interesting but the title does not clearly present the topic of this work. Several issues need to be addresed towards the improvement of this work. The title needs to be improved/rephrased. The abstract is really short and not accurate (it is not clear what is the topic/investigated in this work for example wood to bark adhesion in production of composite panels or in standing trees). You rather use "wood to bark", instead of wood/bark, since it is not clear to the reader (this symbol (/) could be confused with the meaning of "or"). In key words, the phrase "adhesion wood/bark" could not stand alone as a word in my opinion. In line 29, you rather keep it like [3-6]. The introduction is well prepared but more bibliography is necessary to be provided in order to describe adequately the state-of-the-art concerning the use of bark. I would recommend to the authors to incorporate in the introduction significant information from the relevant work DOI: 10.1002/bbb.2291. The scientific, latin names should be in italics. In references brackets, you should start with the lower number of reference. It is not clear why you refer “dormant and 105 vegetation period” in the text, how this factor plays a role to the adhesion. In figure 1, please provide the standard deviation values as well as in all the graphs. Why did you choose to present the graph in figure 1, in introduction and not in the results? You should provide more information about the raw materials, how many trunks etc. and which were the standards applied for shear strength, where did you base the choice of dimensions of the speciemens. The materials and methods section does not describe the statistical analysis applied. Please provide a configuration of the specimens with dimensions on it since it would be extremely useful to the reader. Where such materials of wood-bark could be used? Conclusions are too extended and need to be shortened and more accurate.

Author Response

(The authors gave the same response as above.)

Reviewer 3 Report

The manuscript deals with the wood/bark adhesion of some European commercial tree species, tested by static shear test in axial and tangential wood anatomical direction. The manuscript has typical IMRAD (Introduction / Material and Methods / Result and Discussion) structure. The introduction is wide and deep enough, however a part of the content is unclear, and not fully understandable. The methodology, which is traditional has to be in a part more precisely defined. The results and discussion are well presented, however some details with needed additional information or even testing, need to be improved. The conclusions are to wide and are not full supported by the findings.

Comments / Remarks:

P1 L1 The title of the manuscript should be changed (wording)! Please remove "the effect of different wood species" out if it. Usually it is not used term "effect of wood specie on something".

P1 L15 "Presented article describes testing of the adhesion of wood / bark..." The term adhesion is typically used for action of adhering, which defines molecular attraction exerted between the surfaces of bodies in contact. Please consider of using some other methodology. Isn't it possible to define this as a shear strength of cambial zone?

P2 L56 "According to (ref) cambium can remarkably influence the activity of different mechanical properties..." Wording?!? The sentence has no meaning at the moment!

P2 L72 There is no reference in the text for Figure 1 before its position. Please add it after referencing Add as well the information, on the direction of loading and moisture content state!

Material and methods

P3 L118 Please define at what moisture content, the samples were tested! Did you test green samples, or conditioned samples in some climate?

P4 L145 Is the shear test related to any standard? If, then also the data on shear modulus should be available?!? Regarding the test, please add (or show in the results) a typical stress strain curve.

Results

P5 L159 "These values indicate a remarkable effect of cell placement as well as the structure of the layers in its nearness,..."

Please explain in the results, if the stress, especially at max. load was totally shear? Due to very low values of stress, and know low compression strength, especially in transverse direction, it might be present crushing of samples due to override of compression strength. It is suggested to add to the results some information on typical fracturing behavior of tested samples of used wood species. This would be the way in support of the above statement. Some microscopy / 3D CT scan data of fractures would help a lot.

P6 L193 "Therefore, we can assume that phloem fibers under load will behave the same as mechanical tissues in wood,..."

It is not clear, what do you mean with term "mechanical tissues"? You are actually discussing on mechanical response to the load of three, anatomically and structurally very different wood species.

P6 L203 "The application of analysis..., namely the wood species that remarkably affects the value of shear strength (Figure 6)"

Please add into the sentence, as well as to the Figure 6, the information on direction of loading.

P7 211 "The reason of different shear strength may be the different structure of the bark and the activity of the cambium."

The term "activity of cambium" is usually reserved for the physiological processes in trees. It is much related with the cell division. Please rewrite the sentence to be more understandable - did you have in mind the thickness and number of weak cambium cells? Please correct the same information also in conclussions - P8 L240

Conclusions

P8 L248... From this line several statements and comments is given, to explain the shear strength of wood / bark layer of tested species, as well as used directions of loading. However, the statements have no support in the results, given. Since these are just assumptions, they are redundant.

Non-supported statements:

P8 L248 "These probably, like mechanical tissues in wood, show similar behavior,..."

P8 L256 - L263 "This results can be explained..."

P8 L264 - L269 "In the final evaluation,..., but also by other factors, such as anisotropy..."

Author Response

(The authors gave the same response as above.)

Round 2

Reviewer 1 Report

It's a good article, printable.

Though I might have more recent literature.

Reviewer 2 Report

As I have checked the authors have implemented most of the proposed changes in the revised verion of manuscript towards the improvement of their work. Almost all the changes have been implemented and in my opinion, the manuscript is quite well-prepared and organized enough in the present form.
Nevertheless, figure 4 needs to be clarified and improved to be readable. Additionally, something is going wrong with the standard deviation values presented in the graphs. Further improvement and a general check should be implemented for syntactical and type errors in the whole text.
I remain at your disposal for any clarification.

Reviewer 3 Report

Th version of the manuscript after the first revision is significantly improved. Many of the comments and remarks were successfully considered, and included in the rewritting and reorganizing of the manuscript. Before the continuation of the publishing process, it is suggested to exclude the 6th connclusion, since it does not have the support in the findings:

" Based on the analysis of variance (ANOVA),...,. These results can be explained mainly by different structure of the bark between tested wood species. Mainly differences..."

The above sentence have no support in the findings! There are not findings of submitted study!